# The Urinary Resistome of Clinically Healthy Companion Dogs: Potential One Health Implications

**DOI:** 10.3390/antibiotics11060780

**Published:** 2022-06-08

**Authors:** Tonatiuh Melgarejo, Nathan Sharp, Janina A. Krumbeck, Guangxi Wu, Young J. Kim, Annika Linde

**Affiliations:** 1Veterinary Clinical Center, College of Veterinary Medicine, Western University of Health Sciences, 611 E. Second Street, Pomona, CA 91766, USA; nsharp@westernu.edu (N.S.); yjkim@westernu.edu (Y.J.K.); 2MiDOG LLC, 14672 Bentley Cir, Tustin, CA 92780, USA; jkrumbeck@zymoresearch.com (J.A.K.); gwu@zymoresearch.com (G.W.)

**Keywords:** antimicrobial resistance, dog, urine, microbiome, resistome, species-spanning

## Abstract

An interdisciplinary approach to antimicrobial resistance (AMR) is essential to effectively address what is projected to soon become a public health disaster. Veterinary medicine accounts for a majority of antimicrobial use, and mainly in support of industrial food animal production (IFAP), which has significant exposure implications for human and nonhuman animals. Companion dogs live in close proximity to humans and share environmental exposures, including food sources. This study aimed to elucidate the AMR-gene presence in microorganisms recovered from urine from clinically healthy dogs to highlight public health considerations in the context of a species-spanning framework. Urine was collected through cystocentesis from 50 companion dogs in Southern California, and microbial DNA was analyzed using next-generation sequencing. Thirteen AMR genes in urine from 48% of the dogs {*n*=24} were detected. The most common AMR genes were aph(3′)Ia, and ermB, which confer resistance to aminoglycosides and MLS (macrolides, lincosamides, streptogramins) antibiotics, respectively. Antibiotic-resistance profiles based on the AMR genes detected, and the intrinsic resistance profiles of bacterial species, were inferred in 24% of the samples {*n*=12} for 57 species, with most belonging to *Streptococcus*, *Staphylococcus*, and *Corynebacterium* genera. The presence of AMR genes that confer resistance to medically important antibiotics suggests that dogs may serve as reservoirs of clinically relevant resistomes, which is likely rooted in excessive IFAP antimicrobial use.

## 1. Introduction

Antimicrobial resistance (AMR) constitutes a major global health burden, with an annual death toll approaching 750,000, which is expected to reach 10 million by 2050 [1]. Morbidity and mortality from AMR account for approximately 3 million cases and 40,000 deaths annually in the United States (US), and the Centers for Disease Control and Prevention (CDC) define AMR as a national health crisis [2]. In this post-antibiotic era, authorities encourage the rethinking of how societies address AMR. Simply relying on additional antibiotic development is a flawed strategy because the pipeline is running dry, with limited influx of capital for research and development. If essentially all antibiotics become ineffective at combatting infections, the projected cumulative cost to the global economic output is USD 100 trillion [1]. Human health concerns are front and center; however, recognizing AMR as a species-spanning issue is key if the goal is the identification of sustainable measures to combat this threat. Antimicrobial stewardship necessitates an interprofessional approach since veterinary applications account for 80% of antibiotic usage [3]. Waste from factory farms is released into the environment untreated, which is a health hazard to lifeforms that become exposed to environmental antibiotic residues and other pollutants. Moreover, individuals who are vulnerable to environmental contamination may serve as AMR reservoirs [4]. Limited attention has been paid to AMR in companion animals, including dogs, who commonly live as integrated family members. The occurrence of multidrug-resistant bacteria in cultures has been reported in urine from clinically healthy dogs and from dogs with cystitis [5]. Furthermore, AMR in dogs with urinary-tract infections has increased significantly in the last two decades [6]. Additional insights into the patterns of AMR in companion dogs are relevant because they constitute an important therapeutic limitation in veterinary medicine and a risk factor for AMR spillover. Precautions are taken in the handling of specimens from dogs with clinical disease; however, clients and clinicians may be less attentive in reference to clinically healthy individuals. Pathogens are readily detected in the urine microbiomes of clinically healthy dogs with negative urine cultures [7]. Consequently, insufficient hygiene measures could increase the risk of microbial AMR transfer to people. Insight into microbial resistomes is of interest to counteract AMR. The aim of this study was to obtain a preliminary assessment of the prevalence of the urinary AMR in healthy companion dogs. The presence of AMR bacteria that exist as commensals in dogs without causing disease is of interest because these microbes may serve as AMR-gene reservoirs for bacteria with pathogenic potential for humans and dogs.

## 2. Results

Thirteen AMR genes were detected in urine bacteriome samples from 24 dogs (Figure 1). Notably, four samples contained three AMR genes that confer resistance to aminoglycosides, macrolides, lincosamides, streptogramins, tetracycline, and sulfonamide antibiotics (Figure 1).

The most common AMR genes were the aminoglycoside phosphotransferase gene (aph(3′)Ia), which confers resistance to aminoglycosides (11 samples), and the ermB gene, which confers resistance to MLS (macrolides, lincosamides, streptogramins) antibiotics (7 samples), as shown in Table 1. Six different AMR genes were found to confer resistance to MLS, while two different AMR genes were found to confer resistance to aminoglycosides. Thirteen samples contained AMR genes conferring resistance to aminoglycosides, and fourteen samples to MLS (Table 1).

The antibiotic-resistance profiles based on the AMR genes detected and the intrinsic resistance profiles of bacterial species were inferred in 12 samples for 57 species (Table 2), with most belonging to *Streptococcus* (9), *Staphylococcus* (8), and *Corynebacterium* (7) genera. The most intrinsic resistances were detected for nalidixic acid (42), neomycin, amikacin, and gentamicin (15). *Stenotrophomonas maltophilia*, an MDR pathogen linked to serious human infections, was identified in urine from two dogs (this pathogen displayed intrinsic resistance to 20 antibiotics), plus one AMR gene that conferred resistance against sulfonamide in one of the dogs.

## 3. Discussion

The world has relied on antibiotics to treat infections since the discovery of penicillin in 1929. A century later, rather than overcoming infectious diseases, antibiotic resistances have risen to become amongst the most serious global health threats [8]. Bacteria possess an impressive ability to effectively respond to assaults on their microenvironment. These resistances to antimicrobials are based on both intrinsic and acquired resistances. While intrinsic resistances are innate to a bacterial species on the basis of its inherent structural or functional characteristics (i.e., lack of the antimicrobial target, innate efflux pumps, drug inactivation/degradation) and are passed on vertically, the focus of this discussion is the evolutionary pressures (i.e., injudicious use of antimicrobials) that lead to an increase in genetically acquired resistances in bacterial populations. When exposed to antibiotics, many bacteria use cellular mechanisms, such as horizontal gene transfer (conjugation, transformation, transduction) and mobile genetic elements (plasmids, transposons, insertion sequences, integrons, integrative–conjugative elements), to adapt and survive [9]. It is through this ability that bacteria have developed resistance to different antibiotics. The development of resistomes has occurred as a consequence of bacteria’s ability to acquire these genes from related genera and across bacterial taxa (phylum, class, order, family). The term “resistome” was coined in 2006, and there have been multiple studies describing the presence of resistance genes in diverse environments [10]. However, dogs have received limited attention as reservoirs of clinically relevant resistomes. American households are home to an estimated 77 million dogs and 58 million cats, which makes the United States the most companion-animal-oriented society in the world [11]. This is the first report on the urinary resistome in clinically healthy dogs. The term “Clinically healthy dogs” is defined here as a complete state of physical and behavioral well-being based on a client-provided history and physical-examination findings.

Nearly half of the examined dogs had AMR genes conferring resistances to both aminoglycosides and macrolides, which are defined as critically important antibiotics (Figure 1) [12]. The AMR genes with a high occurrence (26%) in the urine of healthy companion dogs included aph(3′)Ia and ant(4′)Ib. These genes confer resistance to aminoglycosides, including amikacin, gentamicin B, kanamycin, neomycin, and paromycin [13]. According to the report “Antimicrobials sold or used in Food Producing Animals” by the United States Food and Drug Administration (FDA) Center for Veterinary Medicine, more than three hundred metric tons of aminoglycosides were used exclusively for IFAP in the United States in 2019 [14]. In small-animal (canine/feline) medicine, the use of aminoglycosides is very limited, as it was made less relevant more than three decades ago by the development of fluoroquinolones, which have superior safety profiles compared to aminoglycosides [15]. Additionally, aminoglycosides have well-established adverse effects in dogs and cats, such as nephrotoxicity and ototoxicity [15]. Aminoglycosides are occasionally used in dogs, but they are typically reserved to treat serious infections caused by aerobic Gram-negative bacteria and staphylococci [15]. Interestingly, none of the dogs in this study received aminoglycosides, or any other antibiotics, for at least twelve months before the urine collection.

The occurrence of MLS phenotype genes was documented in the bacterial urine profiles of fourteen dogs (28%). These AMR genes confer resistances to critically important antibiotics, such as erythromycin and its synthetic derivatives, azithromycin and clarithromycin. Azithromycin is a highly clinically relevant antibiotic for the small-animal internist, as it is often used in canine medicine to treat a variety of bacterial, rickettsial, and parasitic infections [15]. In human medicine, azithromycin and clarithromycin are commonly used in the treatment of a variety of infections, including community-acquired respiratory-tract infections and mycobacterial infections [16]. Public health concerns about the presence of MLS genes in the urine of healthy companion dogs have not been previously addressed. However, they warrant serious consideration because azithromycin is the second most commonly prescribed outpatient antibiotic for humans in the United States [17], and is a critically important antibiotic in human medicine, according to the WHO’s latest antibiotic classification [18]. Importantly, almost five hundred metric tons of macrolides were used exclusively for IFAP in the United States in 2019 [14]. In other words, about half a kiloton of macrolides administered to production animals—mainly (>90%) through water and feed [14]—may have also promoted AMR development via indirect exposure to humans and other animals alike (Figure 2).

The TetC gene confers resistance to tetracycline and its synthetic derivatives, and it was found in 2% of the clinically healthy dogs in this study. Tetracyclines constitute the most frequently sold class (>80%) of medically important antibiotics for use in livestock and are used mainly for cattle and swine production, which accounted for 67% of all sales in the United States in 2019 [14]. According to the Centers for Disease Control and Prevention (CDC), doxycycline (synthetic tetracycline derivative) was prescribed 19.5 million times for human use in 2020, making it the fifth most commonly used oral antimicrobial in outpatients in the United States [17]. The sheer amount of tetracyclines (>9 million pounds in 2019) used for IFAP in the United States would strongly suggest that a significant amount of antimicrobial exposure in people happens through paths other than the intake of prescription drugs from human healthcare providers (Figure 2).

The presence of the blaZ gene was detected in 4% of the dogs in this study. This AMR gene produces an enzymatic inactivation of the penam-family molecules to confer drug resistance to beta-lactams, including critically important antibiotics, such as amoxicillin, ampicillin, cephalexin, dicloxacillin, methicillin, and penicillin [18]. The finding of this gene in the urine of clinically healthy companion dogs is significant because blaZ may confer resistance to at least two (ampicillin and cephalexin) of the top five most prescribed outpatient oral antibiotics in the United States [17]. According to the most recent FDA report (2019) [14], more than seven hundred metric tons of penicillins are used for IFAP in the United States, accounting for 12% of the total sales of antibiotics that are specifically destined for animal agriculture. Notably, 66% of the penicillins used in the United States in 2019 were used in turkey production [14], a species that most Americans customarily consume during the Thanksgiving holiday.

The gyrA gene was found in “only” 2% of the study dogs, but this is nevertheless a noteworthy finding because it is a medically relevant antimicrobial gene that confers resistance to quinolones, likely by point mutations in the bacterial gyrA gene [19]. The fluoroquinolones approved for human use in the United States include ciprofloxacin, gemifloxacin, levofloxacin, moxifloxacin, norfloxacin, and ofloxacin [20]. These wide-spectrum antimicrobial agents are used to treat highly prevalent infections, such as urinary-tract infections (UTIs), joint and bone infections, and soft tissue and skin infections [20]. Disconcertingly, fluoroquinolones might also be one of the limited treatment choices for bacterial warfare agents, such as *Bacillus anthracis, Francisella tularensis,* and *Yersinia pestis* [21]. Even though the annual consumption of fluoroquinolones in IFAP in the United States is relatively low (24 metric tons), these antibiotics are used mainly in cattle and swine production, where most of the antibiotics are administered through the water and feed [14], which thereby constitute a major risk for the release of antimicrobial agents into the environment.

Antibiotic-resistant bacteria from production-animal origin have been considered a key player in the occurrence of resistance to antibiotics for more than three decades [22,23]. Numerous antimicrobials used in IFAP are identical or closely related to the antimicrobials used in humans, and many of these drugs are used explicitly for disease prevention and growth promotion in the absence of clinical disease [24]. Antimicrobial use in the context of IFAP can lead to the selection and dissemination of antimicrobial-resistant bacteria in food-producing animals, which may be transmitted to humans and companion animals via food and other transmission routes [25] (Figure 2). There is clear documentation showing a direct link between AMR and antibiotic use in IFAP [26,27,28]. CDC director Robert R. Redfield, M.D., mentions, in the 2019 report on AMR, that this situation is alarming [29], and he states that the post-antibiotic era is here, considering that nearly 3 million people across the country are facing an antibiotic-resistant infection, which is costing USD 65 billion a year. The judicious use of antimicrobials in production animals remains a prerequisite to delay the emergence of resistance to the still-working antibiotics. Despite international efforts and global resolutions from many international agencies, including the World Health Organization (WHO), the Food and Agriculture Organization of the United Nations (FAO), and the World Organization for Animal Health (OIE) [24], the future appears bleak if IFAP continue with business as usual and without a major modification to antibiotic-use practices. It has been projected that global antimicrobial consumption will rise by 67% by 2030, almost doubling the use in South America (including Brazil), China, India, and Russia [3]. This increase will likely be due to the growth in consumer demand for animal products in middle income countries, and the move to factory farming involving excessive antimicrobial use.

An objective limitation of any study investigating the resistome of an ecological habitat is that it could never cover all the types of different antimicrobial resistances. As the scientific community continues to investigate the different mechanisms of antimicrobial resistances in the bacterial kingdom, previously unreported mechanisms may be discovered, as well as previously unreported antimicrobial-resistance genes. Any antimicrobial-resistance-gene panel will therefore always be limited and will not be able to cover 100% of all the possible antimicrobial resistances in any given habitat. Another limitation of this study is the sample size of 50 dogs. These dogs provided an opportunistic sample for this study, as they were previously investigated for their bacterial and fungal microbiomes [7]. Future studies may include a larger sample cohort from a more diverse geographical and medical background. Ideally, studies such as this could be extended to other companion-animal species (i.e., cats, birds, and rodents) to further monitor the development of the bacterial resistome in companion animals.

In summary, this study detected 13 different AMR genes in the urine of 48% of clinically healthy companion dogs, which was rather surprising considering that: (1) the urine cultures from all the dogs rendered negative results; (2) the dogs received no antibiotics for at least 12 months prior to urine collection; and (3) all the dogs remained healthy at the 18-month follow-up exams. The presence of AMR genes in dogs has clinically relevant implications for companion-animal and human health.

## 4. Materials and Methods

### 4.1. Subjects included, Urine Sample Collection, and Analysis of Urine (UA)

Clinically healthy companion dogs (*n* = 50) were prospectively identified at the Western University of Health Sciences, Pomona, CA, USA (WesternU). “Clinically healthy” dogs were defined as ADD HERE. This study was approved by the Institutional Animal Care and Use Committee (protocol # R19IACUC037)). Dogs’ signalment, medical history, and physical examination were recorded when dogs were admitted to the Pet Health Center at WesternU. A standardized protocol was used to minimize variability in urine collection, as described before [7]. Briefly, a single urine sample (6 mL) was collected via ultrasound-guided cystocentesis, with each dog in dorsal recumbency. Each urine sample was divided as follows: 2 mL into a BD Vacutainer Urinalysis tube (Becton, Dickinson and Company, Franklin Lakes, New Jersey) for standard urinalysis, and 4 mL into a MiDOG Urine Collection tube (MiDOG LLC, Tustin, California) for urinalysis. Collection tubes contain 50 μL of urine conditioning buffer (Cat. No. D3061-1-140, Zymo Research Corp., Irvine, California), which preserves the microbial profile for prolonged periods at ambient temperature. All urine samples were batched and stored at 4C until processing. Urine samples were subsequently analyzed by MiDOG LLC, as described before (7). A complete urinalysis (UA) was performed immediately after sample collection and included visual exam, dipstick test, microscopic exam, and microbiology (aerobic culture and susceptibility). All urine samples were kept at 4 °C and processed the same day that urine was collected. Inclusion criteria included clinically healthy dogs without a history of antibiotic treatment for the preceding twelve months, current on vaccinations, spayed/ neutered, a body condition score (BCS) between 5 and 6, and negative aerobic urine culture results. Exclusion criteria included dogs treated with antibiotics in the past year, intact, not up to date on vaccinations, a BCS below 5/9 or above 6/9, or positive urine culture.

### 4.2. DNA Extraction and Urine Microbiota 16S Analysis

The methodology applied here was previously described [7]. Briefly, genomic DNA was purified using the ZymoBIOMICS-96 DNA kit (Cat. No. D4304, Zymo Research Corp.), according to the manufacturer’s instructions, in conjunction with a Hamilton Star liquid-handling robot (Hamilton Company, Reno, Nevada). Zymo Research Corp. performed the sample library preparation and data analysis for both bacterial and fungal profiling (also the manufacturer of subsequent catalogue numbers, unless otherwise stated). Libraries were prepared using the Quick-16S NGS Library Prep Kit (Cat. No. D6400), according to the manufacturer’s instructions, with minor modifications. Primer sequences target the V1-to-V3 region of the 16S rDNA, as previously described [7]. The exact sequences are proprietary to the MiDOG LLC service (https://www.midogtest.com, accessed on 13 November 2021). Libraries were sequenced using an Illumina HiSeq 1500 (Illumina, San Diego, CA, USA). Sequence reads were quality controlled using Dada2 (R package version 3.4.9). Reads were trimmed and paired-end reads were merged (perfect matching to primers, and no ambiguous nucleotides were allowed). The microbiota profile of each sequenced urine sample was determined using the bioinformatics analysis pipeline offered by the MiDOG LLC testing service, which provides amplicon-sequence-variant (ASV)-level (roughly species-level) taxonomic identification. All phylotypes were computed as percent proportions on the basis of the total number of sequences in each sample. To control for any potential contamination of sequencing buffers, equipment, and other material, several negative controls were run for the extraction process, as well as the library preparation.

### 4.3. Bioinformatics and Statistical Analysis

The methods applied here have been previously described [7]. Briefly, Shapiro–Wilks tests were used to test for the normality of the data. Data tables generated by the MiDOG analytical pipeline were used for descriptive summaries and statistical hypothesis testing using the R software package. The number of reads for each ASV were summed for all of the negative controls, and this total was subtracted from each of the samples. In cases where this produced a negative number, the number of reads was recorded as 0. For initial descriptive statistics, bacterial genera with fewer than 10 reads were removed from the analysis.

### 4.4. Detection of the Urinary Resistome

The presence of AMR genes in the urine of healthy companion dogs was evaluated by using a proprietary sequencing workflow that targets at least eighty AMR genes. An amplicon-based sequencing approach was applied using proprietary PCR primers, which were designed on the basis of AMR-gene sequences retrieved from NCBI (National Center for Biotechnology Information). Sequencing reads were mapped back to the reference using a proprietary pipeline. To ensure specificity and reproducibility of the tests, sequencing reads were further confirmed using the Comprehensive Antibiotic Resistance Database (CARD) [30].

## 5. Conclusions

Human exposure to canine resistomes is a public health concern that warrants further investigation (Figure 2), and the clinical relevance of our findings requires dedicated interdisciplinary exploration. While the study is inherently limited by its focus on the incidence of AMR genes in urine from clinically healthy companion dogs, one might postulate what the potential clinical implications are for dogs, and for humans coexisting with these dogs. The presence of AMR genes is likely to translate into limited or no efficacy for specific antibiotics that are used in the context of treating infections in either the dogs, or in humans living in close proximity to the dogs. More vulnerable populations of dogs and/or humans would be composed of individuals who are very young, elderly, or immunocompromised. This is especially important given that AMR is a global health crisis that caused nearly five million deaths from diseases associated with bacterial AMR in 2019, including more than one million deaths directly attributable to bacterial AMR [31]. Furthermore, AMR killed more people than HIV and malaria combined in 2019 [31]. Anthropogenic activities shape environmental resistomes and impact the allostatic loads in humans and dogs. Excessive antibiotic use in IFAP now presents as a conceivable root cause of increased resistome frequencies, and healthy dogs living in human societies have seemingly become innocent bystanders [4].

## Figures and Tables

**Figure 1 antibiotics-11-00780-f001:**
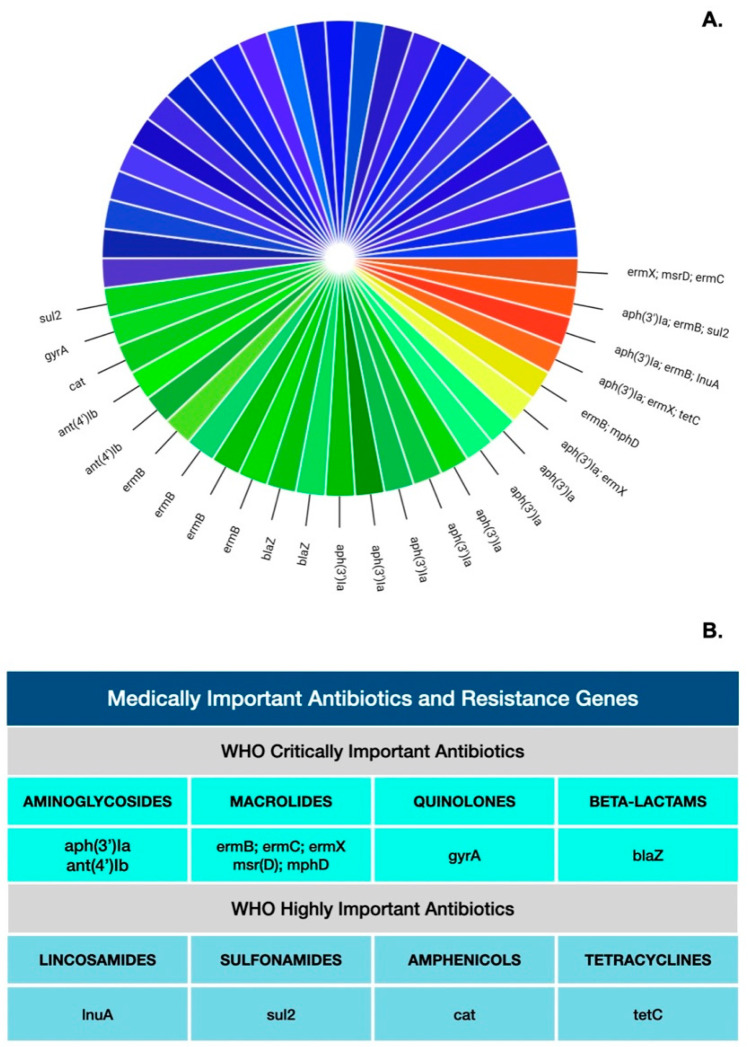
AMR gene presence in urine microbiome samples from clinically healthy dogs. The pie-chart slices represent individual dogs (*n* = 50) included in the study (**A**). Slices in gradients of red, yellow, and green represent samples containing 3, 2, and 1 AMR genes, respectively. Samples shown in gradients of blue were negative for all tested AMR genes. All 13 AMR genes detected in samples from 24 dogs confer resistance against antibiotics defined by the WHO as medically important (**B**).

**Figure 2 antibiotics-11-00780-f002:**
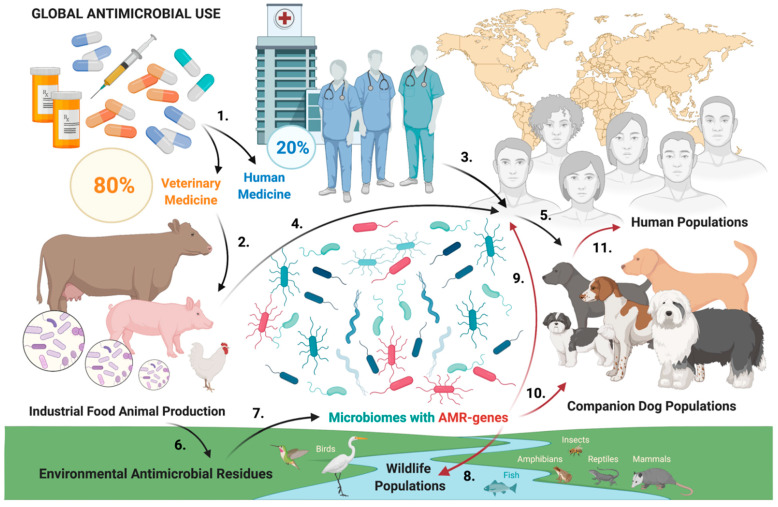
Public health exposure paths in global antimicrobial use. Total antimicrobial use is distributed approximately 80% vs. 20% between veterinary and human medicine, respectively (1). A majority of veterinary antimicrobial use (2) is allocated towards industrial food animal production (IFAP), which contains an estimated 99% of all farmed animals. A significant amount of antimicrobial exposure in people therefore happens through paths other than the intake of prescription drugs from human healthcare providers (3). Consumption of animal-derived products from IFAP also entails ingestion of antimicrobial residues for both humans (4) and their companion dogs (5). Dogs are also prescribed antimicrobials from veterinary clinicians, but this use is minimal compared to IFAP. Waste from IFAP, moreover, results in the release of antimicrobial residues (6) and other contaminants into the environment. Excessive antimicrobial use and environmental antimicrobial residues fuel the development of microbiomes with varying levels of resistances (7), at an alarming rate and scale. The resulting antimicrobial resistance (AMR) constitutes a health hazard for wildlife (8), humans (9), and companion-dog populations (10). The presence of microbiomes with AMR genes in dogs, including a canine urinary resistome, comprise an additional risk factor that could spill over into human populations (11). Created with BioRender.com.

**Table 1 antibiotics-11-00780-t001:** Overview of antibiotics that the detected AMR genes confer resistance to, as well as enzymes involved in conferring resistance, with total number of positive urine samples collected from 50 clinically healthy dogs (percentage of positive samples is included in parenthesis). Beta-lactam antibiotics cover penicillins, cephalosporins, and related compounds. The targeted AMR genes could confer resistance to 44 antibiotics evaluated in the workflow, including Amikacin, Amoxicillin, Ampicillin, Azithromycin, Cefadroxil, Cefazolin, Cefovecin, Cefoxitin, Cefpodoxime, Ceftazidime, Ceftiofur, Cephalexin, Cephalothin, Chloramphenicol, Clavamox, Clindamycin, Ciprofloxacin, Colistin, Doxycycline, Enrofloxacin, Florfenicol, Gentamicin, Imipenem, Levofloxacin, Lincomycin, Marbofloxacin, Metronidazole, Minocycline, Mupirocin, Neomycin, Nitrofurantoin, Orbifloxacin, Oxacillin, Penicillin, Penicillin G, Piperacillin, Pradofloxacin, Rifampin, Sulfonamide, Tetracycline, Timentin, Ticarcillin, Tobramycin, and Trimethoprim.

Antibiotics	AMR Gene	Enzyme Conferring AMR	Positives (%)
Aminoglycosides	aph(3′)Ia	aminoglycoside.phosphotransferase	11 (22%)
	ant(4′)Ib	kanamycin.nucleotidyltransferase	2 (4%)
MLS: Macrolides Lincosamides Streptogramins	ermB	23S.ribosomal.rna.methyltransferase	7 (14%)
	ermX	23S.ribosomal.rna.methyltransferase	3 (6%)
	mphD	Macrolide.2.phosphotransferase	1 (2%)
	msr(D)	ABC.F.ribosomal.protection.protein	1 (2%)
	ermC	23S.ribosomal.rna.methyltransferase	1 (2%)
	lnuA	lincosamide.nucleotidyltransferase	1 (2%)
Beta-lactams	blaZ	staph.blaZ	2 (4%)
Sulfonamides	sul2	sulfonamide.resistant.dihydropteroate.synthase	2 (4%)
Amphenicols	cat	chloramphenicol.acetyltransferase	1 (2%)
Quinolones	gyrA	staphylococcus.pseudintermedius.250TTG	1 (2%)
Tetracyclines	tetC	tetracycline.efflux.pump	1 (2%)

**Table 2 antibiotics-11-00780-t002:** Antimicrobial-resistance genes detected in canine urine samples. Bacteria with intrinsic resistances (inferred) to antimicrobials were present in samples from twelve of the dogs (24%), while bacteria carrying AMR genes were detected in seven of these dogs (14%). We found no correlation between AMR-gene presence and the relative abundances of different microbial species.

Bacterial Species (Relative Abundances in Urine Samples)	Genes Detected Against	Intrinsic Resistances Against
*Bifidobacterium longum* (3.1%); *Escherichia-Shigella coli* (6.5%); *Lachnoclostridium* spp. (1.5%); *Rothia mucilaginosa* (5.4%)	Clindamycin, Lincomycin, Erythromycin, Azithromycin	
*Campylobacter upsaliensis* (0.1%)		Clindamycin
*Corynebacterium auriscanis* (45.6%); *C. genitalium* (4.1%); *C. mucifaciens* (26.9%); *C. pilbarense* (9.6%); *C. simulans* (1.7%); *C. pseudogenitalium-tuberculostearicum* (0.7%, 3.0%, 4.2%); *C. ureicelerivorans* (5.9%); *Lactobacillus aviarius* (2.2%); *L. fabifermentans* (0.1%); *L. vaginalis* (1.6%); *Staphylococcus cohnii* (1.1%); *S. capitis-caprae* (1.1%); *S. epidermidis* (1.3%, 2.5%); *S. felis* (4.6%); *S. hominis* (8.3%)		Nalidixic acid
*Corynebacterium pseudogenitalium-tuberculostearicum* (4.2%)	Clindamycin, Lincomycin, Erythromycin, Azithromycin, Neomycin, Amikacin, Gentamicin	Nalidixic acid
*Corynebacterium ureicelerivorans* (1.9%); *Enterococcus cecorum* (2.06%); *Lactobacillus johnsonii* (1.6%); *L. salivarius* (4.5%); *Staphylococcus epidermidis* (0.5%); *S. hominis* (0.3%)	Clindamycin, Lincomycin, Erythromycin, Azithromycin	Nalidixic acid
*Escherichia-Shigella boydii-coli-fergusonii* (0.9%)	Neomycin, Amikacin, Gentamicin	
*Haemophilus parainfluenzae* (0.6%)		Clindamycin
*Pseudomonas* spp. (4.0%, 7.1%, 8.2%, 14.4%, 20.4%); *P. moraviensis* (0.2%)		Cefadroxil, Cefazolin, Cefoxitin, Clindamycin, Erythromycin, Azithromycin, Rifampicin
*Staphylococcus hominis* (3.8%); *S. simulans* (6.7%); *S. pseudintermedius* (17.8%); *S. delphini-intermedius-pseudintermedius* (100.0%)	Ampicillin, Amoxicillin, Oxacillin, Benzylpenicillin	Nalidixic acid
*Stenotrophomonas maltophilia* (3.0%) *Stenotrophomonas maltophilia* (5.7%)	Sulfonamide	Ampicillin, Amoxicillin, Cefadroxil, Cefazolin, Cefpodoxime, Ceftiofur, Cefovecin, Cefoxitin, Clindamycin, Erythromycin, Azithromycin, Doxycycline, Tetracycline, Neomycin, Amikacin, Gentamicin, Clavamox, Piperacillin/Tazobactam, Imipenem, Rifampicin
*Streptococcus alactolyticus* (1.4%); *S. canis* (2.6%, 44.2%); *S. gordonii* (1.6%); *S. oralis* (2.0%); *S. parasanguinis* (16.8%); *S. salivarius* (0.4%, 6.7%); *S. salivarius-thermophilus* (1.1%); *S. sanguinis* (0.4%, 3.4%)		Neomycin, Amikacin, Gentamicin, Nalidixic acid
*Streptococcus cristatus* (1.3%); *S. sanguinis* (1.3%)	Clindamycin, Lincomycin, Erythromycin, Azithromycin,	Neomycin, Amikacin, Gentamicin, Nalidixic acid

## Data Availability

Third-party data restrictions apply to the availability of these data. Data were obtained from MiDOG LLC, and the data include proprietary information regarding the sequencing specifics. Data for the abundance tables are available from the corresponding author (T Melgarejo) after written permission from MiDOG LLC has been granted, and upon reasonable request.

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
