# Peer review of "The Urinary Resistome of Clinically Healthy Companion Dogs: Potential One Health Implications"

_antibiotics, 2022, doi:10.3390/antibiotics11060780_

Round 1

Reviewer 1 Report

The manuscript by Melgarejo et al. is undoubtedly original with important contribution to the understanding of the antimicrobial gene presence in urine from clinically healthy dogs. I support its possible further acceptance after appropriate modifications as outlined below:

 Line 24: please ensure that the scientific name of all of the mentioned genera and species are italicized throughout the manuscript;

Line 17: „AMR -gene presence in microorganisms recovered from urine” instead of „AMR-gene presence in urine” – please take into consideration that AMR-gene can be harbored only bacteria and not urine. Please take into consideration this concern throughout the manuscript.

Line 34: „US, and the CDC” – please define these acronyms being their first appearance in the text

Line 38: please replace the term „antibiotics” with „antimicrobials”  throughout the manuscript

Line 48: „multi-drug resistant” for this expression, please use the „MDR” acronym

Line 58: „This is the first report on the urinary resistome in clinically healthy dogs.” This sentence is fit better to the discussion chapter. Please clearly define at the end of the introduction chapter the study aim.

Line 63: „Thirteen” instead of „Thirteen (13)”

Line 65: „MLS (macrolides, lincosamides, streptogramins)” – in this form, the acronym using is not appropriate

Line 92: in the last column of the Table 1, please insert the value of the 95% confidence interval for the reported percentages

Line 106 – table 2 please express the reported percentages with a single decimal value

Line 243: „WesternU” – please provide further information about the city and country of the establishment

Line 240: the authors must to detail the term „clinically healthy” in the materials and methods section

Line 241: please justify your choice in the total number of included dogs in the study. Why exactly 50 and not more or less? In this regard, the authors need to refer to a statistical model, based on which they can validate the study results. So, the authors must convince the scientific community that they results are completely supportable by statistical tools, without any doubt of speculation. This is very important and can be considered a major weakness of this study!

Line 302: the first two sentences from the conclusion sections must to be deleted, they are not appropriate. The authors must clearly highligt the study limitations and future perspectives

The reference list is not in agreement with the journal requirements.

Author Response

REVIEWER 1:

Comments and Suggestions for Authors

The manuscript by Melgarejo et al. is undoubtedly original with important contribution to the understanding of the antimicrobial gene presence in urine from clinically healthy dogs. I support its possible further acceptance after appropriate modifications as outlined below:

 Line 24: please ensure that the scientific name of all of the mentioned genera and species are italicized throughout the manuscript.

R: This has been updated throughout the manuscript.

Line 17: „AMR -gene presence in microorganisms recovered from urine” instead of „AMR-gene presence in urine” – please take into consideration that AMR-gene can be harbored only bacteria and not urine. Please take into consideration this concern throughout the manuscript.
R: We reworded the abstract accordingly, added it to lines 63, 142

Line 34: „US, and the CDC” – please define these acronyms being their first appearance in the text

R: This has been updated.

Line 38: please replace the term „antibiotics” with „antimicrobials”  throughout the manuscript.

R: This was not done because some of the core the references used for the manuscript (CDC, WHO, FAO) refer specifically to “antibiotics” and not “antimicrobials”. Exchanging these words throughout would therefore be incorrect.

Line 48: „multi-drug resistant” for this expression, please use the „MDR” acronym.

R: Done

Line 58: „This is the first report on the urinary resistome in clinically healthy dogs.” This sentence is fit better to the discussion chapter. Please clearly define at the end of the introduction chapter the study aim.

R: Modified accordingly in the manuscript. The aim of this study was to get a preliminary assessment of the prevalence of AMR in healthy companion dogs.

Line 63: „Thirteen” instead of „Thirteen (13)”
R: This has been updated.

Line 65: „MLS (macrolides, lincosamides, streptogramins)” – in this form, the acronym using is not appropriate
R: This has been updated accordingly.

Line 92: in the last column of the Table 1, please insert the value of the 95% confidence interval for the reported percentages
R: This was a pilot study and hence adding the 95% confidence to the table is not feasible

Line 106 – table 2 please express the reported percentages with a single decimal value R: This has been updated accordingly.

Line 243: „WesternU” – please provide further information about the city and country of the establishment
R: This has been updated accordingly.

Line 240: the authors must to detail the term „clinically healthy” in the materials and methods section.
R: Modified in the manuscript by inserting the following sentence: “Clinically healthy dogs” is here defined as a complete state of physical and behavioral well-being based on a client provided history and physical examination findings.

Line 241: please justify your choice in the total number of included dogs in the study. Why exactly 50 and not more or less? In this regard, the authors need to refer to a statistical model, based on which they can validate the study results. So, the authors must convince the scientific community that they results are completely supportable by statistical tools, without any doubt of speculation. This is very important and can be considered a major weakness of this study!
R: The sample size has been added as a limitation of this manuscript to the discussion.

Line 302: the first two sentences from the conclusion sections must to be deleted, they are not appropriate. The authors must clearly highlight the study limitations and future perspectives
R: Study limitations and future perspectives have been added to the discussion section. Conclusion was modified according to the reviewer’s comments.

The reference list is not in agreement with the journal requirements.

R: References have been checked *

Reviewer 2 Report

This is an interesting approach to filling some gaps in our understanding of the resistome of companion animals. Where the paper sticks to reporting on that, it does quite well. I have a couple of general comments, as well as several specific recommendations for strengthening the manuscript. 

GENERAL: 

1. I do not see the relevance of any of the discussion about antibiotic use in food animals or the references to "industrialized food animal production" and I urge the authors to remove and the editor to insist on removal of those discussions throughout the paper, as they have no relevance to the study reported herein nor its interpretation. 

2. Intrinsic resistance should not be combined with acquired resistance (presence of AMR genes) and then interpreted as of public health or clinical importance. 

Other specific comments: 

3. It would be helpful for the reader if the total number of samples or dogs was included next to the percentages report in the abstract, e.g., at Line 24. It is a striking number until one reads closely and finds it is from 50 dogs from one small region. 

4. Figure 1: It might be easier to read and interpret this figure if the same genes were situated next to each other on the circle. Also, it is important to note that although these antibiotics are of medical importance, the study did not LOOK for other resistance to other antibiotics, so they would not have been detected. 

5. In Table 1, please verify that each gene listed next to MLS confers resistance to all 3 groups of drugs, and if not, consider other ways of splitting them out. Also, in the legend, isn't it true that genes were targeted, not antibiotics? 

6. Line 96: This study did not DETECT intrinsic resistance, it only identified organisms that are intrinsically resistant. This is not the same and should not be confounded. In addition, intrinsic resistance cannot be spread to other organisms, so it is not of the same public health concern nor will antibiotic stewardship activities impact intrinsic resistance. 

7. Table 2: As discussed above, instrinsic resistance should not be compared to acquired resistance/AMR genes or displayed in the same table. These "resistances" were not detected, they are inferred from the organism. 

8. Line 115: Bacteria do not adapt to antibiotics; there is selection pressure, and those able to survive pass on their genes. 

9. Line 133: Citations are needed for the statements about what drugs are used and for which indications. 

10. Limitations of this study should be included in the discussion and/or the conclusions including (1) convenience sample of dogs from one institution and one time period, (2) no evaluation in the study nor discussion in the paper about the actual risk of any of the organisms being able to transfer resistance, especially in the urine environment, (3) cultures were negative so the numbers of organisms are very low and may be of no risk at all of transferring resistance genes (what is the evidence that they might?), (4) phenotype and genotype were not determined together, and genotype doesn't necessarily lead to the phenotype of clinical resistance, (5) the lack of reporting of data on abundance due to its proprietary nature. 

11. Line 258: The more appropriate term is "susceptibility" not "sensitivity". 

12. Line 269: This sentence appears to be missing a subject. 

13. Line 295: Statistics are in the methods but not reported in the paper. 

14. Conclusions: Discuss the relevance to dogs, dog owners, and clinical resistance in dogs or in humans living with dogs, not about food animals or total antibiotics sold, which were not evaluated in nor relevant to this study. 

Author Response

REVIEWER 2:

Comments and Suggestions for Authors

This is an interesting approach to filling some gaps in our understanding of the resistome of companion animals. Where the paper sticks to reporting on that, it does quite well. I have a couple of general comments, as well as several specific recommendations for strengthening the manuscript.

GENERAL:

  1. I do not see the relevance of any of the discussion about antibiotic use in food animals or the references to "industrialized food animal production" and I urge the authors to remove and the editor to insist on removal of those discussions throughout the paper, as they have no relevance to the study reported herein nor its interpretation.

R: In our manuscript we clearly cite that antibiotic-resistant bacteria from production animal origin have been considered a key player in the occurrence of resistance to antibiotics for more than three decades (lines 196 to 204; and line 209 to 217 from the original manuscript). Removing discussion of antibiotic use by Industrialized Food Animal Production (IFAP) would be non-scientific because antibiotic use in veterinary medicine accounts for 80% of global antibiotic use. Therefore, we maintain that removal of this discussion from the manuscript would be entirely against the main scientific message the authors are communicating throughout the paper. For this reason, we are not going to remove all the discussion related to antibiotics and IFAP.

  1. Intrinsic resistance should not be combined with acquired resistance (presence of AMR genes) and then interpreted as of public health or clinical importance.
    R: We recognize the reviewer’s comments that there is a considerable difference between acquired and intrinsic resistance. We have addressed that in the manuscript in the first paragraph of the discussion section.

Other specific comments:

  1. It would be helpful for the reader if the total number of samples or dogs was included next to the percentages report in the abstract, e.g., at Line 24. It is a striking number until one reads closely and finds it is from 50 dogs from one small region.
    R: This has been modified accordingly in the manuscript.

  2. Figure 1: It might be easier to read and interpret this figure if the same genes were situated next to each other on the circle. Also, it is important to note that although these antibiotics are of medical importance, the study did not LOOK for other resistance to other antibiotics, so they would not have been detected.
    R: We did change the order of genes in the as recommended
    While our manuscript focused on the resistances that were detected, we did screen for additional resistances as described in the legend of table 1
    .

  3. In Table 1, please verify that each gene listed next to MLS confers resistance to all 3 groups of drugs, and if not, consider other ways of splitting them out. Also, in the legend, isn't it true that genes were targeted, not antibiotics?
    R: We did verify that all resistance genes and their associated antibiotic resistance (including MLS) are aligned with the CARD (reference 31 of the manuscript)

  1. Line 96: This study did not DETECT intrinsic resistance, it only identified organisms that are intrinsically resistant. This is not the same and should not be confounded. In addition, intrinsic resistance cannot be spread to other organisms, so it is not of the same public health concern nor will antibiotic stewardship activities impact intrinsic resistance.

R: Intrinsic resistance (IR) was reported to facilitate further analysis, and we did not claim that we detected IR. We agree with the reviewer, we have clarified this point modifying the first paragraph of the discussion section.

  1. Table 2: As discussed above, intrinsic resistance should not be compared to acquired resistance/AMR genes or displayed in the same table. These "resistances" were not detected, they are inferred from the organism.
    R: We have clarified the difference between AMR genes and intrinsic resistance in the manuscript by modifying the first paragraph of the discussion, and also by modifying the legend of table 2.

  1. Line 115: Bacteria do not adapt to antibiotics; there is selection pressure, and those able to survive pass on their genes.
    R: Manuscript was modified in the discussion section.

  1. Line 133: Citations are needed for the statements about what drugs are used and for which indications.

R: citation added to the corresponding line

  1. Limitations of this study should be included in the discussion and/or the conclusions including (1) convenience sample of dogs from one institution and one time period, (2) no evaluation in the study nor discussion in the paper about the actual risk of any of the organisms being able to transfer resistance, especially in the urine environment, (3) cultures were negative so the numbers of organisms are very low and may be of no risk at all of transferring resistance genes (what is the evidence that they might?), (4) phenotype and genotype were not determined together, and genotype doesn't necessarily lead to the phenotype of clinical resistance, (5) the lack of reporting of data on abundance due to its proprietary nature.

R: 1) We agree with the reviewer, this was an opportunistic study

2) It was not within the scope of the manuscript to calculate actual risk of transferring resistance between bacteria. The aim was just to report the prevalence of AMR-genes in urine of healthy companion dogs. 3) Bacteria may be viable but unculturable. Additionally, some bacteria are difficult to grow like Ureaplasma, Mycoplasma, which does not necessarily mean they are low in numbers. Also, the transferring of resistance genes was not within the scope of this manuscript. 4) We couldn’t evaluate the phenotype because the bacteria were not culturable. By looking at the genotype, we are reporting the resistance potential of the ecosystem. 5) Yes, the lack of reporting of data on abundance was mainly due to the proprietary nature of the work. Additionally, we were only measuring potential of resistance.  

  1. Line 258: The more appropriate term is "susceptibility" not "sensitivity".
    R: This has been updated.

  1. Line 269: This sentence appears to be missing a subject.
    R: Corrected in the manuscript.

  1. Line 295: Statistics are in the methods but not reported in the paper.
    R: Body text of the statistical methods was modified accordingly.

  2. Conclusions: Discuss the relevance to dogs, dog owners, and clinical resistance in dogs or in humans living with dogs, not about food animals or total antibiotics sold, which were not evaluated in nor relevant to this study.
    R: Conclusion has been modified according to reviewer’s comments.

Reviewer 3 Report

One would have expected simultaneous transmission of AMR and virulence genes, and hence a positive correlation. The strong negative correlation must be explained to a greater detail.

The lack of similarity between profiles human and animal isolates is rather surprising given the fact that E. coli is a One Health pathogen that can be passed between species via the environment. On the same premise, why did the authors excluded samples from the environment?

The choice of antibiotics used in the study must be justified. Was it based on CLSI guidelines or history of antibiotic use in the locality? We know that resistance to the chosen antibiotics is very high worldwide and the portrayed results were to be expected.

In the determination of susceptibility to antimicrobials no standard organisms were involved/mentioned

What is the significance of finding more virulent genes in isolates from animals?

It is mentioned that 21 isolates (15%) were un-type able without offering an explanation.

Author Response

REVIEWER 3:

Comments and Suggestions for Authors

One would have expected simultaneous transmission of AMR and virulence genes, and hence a positive correlation. The strong negative correlation must be explained to a greater detail.

R: Our manuscript did not assess the prevalence of virulence genes of the microbes reported here. Further, we did not report any negative correlation in need for the clarification from the reviewer.

The lack of similarity between profiles human and animal isolates is rather surprising given the fact that E. coli is a One Health pathogen that can be passed between species via the environment. On the same premise, why did the authors excluded samples from the environment?

R: The analysis of human profiles was not within the scope of this work. As seen in table 2, we did report on E. coli specifically. Additionally, the aim of the study was to assess the urinary resistome in clinically healthy dogs, not the environment.

The choice of antibiotics used in the study must be justified. Was it based on CLSI guidelines or history of antibiotic use in the locality? We know that resistance to the chosen antibiotics is very high worldwide and the portrayed results were to be expected.

R: We did not use any antibiotics in this study. The aim of the study was mainly to report the prevalence of AMR-genes in urine of healthy companion dogs using next-generation sequencing.

In the determination of susceptibility to antimicrobials no standard organisms were involved/mentioned.
R: We did not determine any susceptibility. Instead, we inferred antimicrobial resistance based on DNA evidence, rather than culture-based antibiotic sensitivity.

What is the significance of finding more virulent genes in isolates from animals?

R: We did not assess any virulent genes, nor did we report any presence of virulent genes. We reported only the AMR-genes in the urine of clinically healthy companion dogs.  

It is mentioned that 21 isolates (15%) were un-type able without offering an explanation.

R: We do require further explanation from the reviewer on which isolates he/she is referring to. We did not isolate any bacteria but exclusively analyzed urine bacterial DNA. All bacteria that are reported in the manuscript are identified to the species level, so we are not sure on what “untypable’ is referring to.